# Formulation of Chitosan Microparticles for Enhanced Intranasal Macromolecular Compound Delivery: Factors That Influence Particle Size during Ionic Gelation

**DOI:** 10.3390/gels8110686

**Published:** 2022-10-23

**Authors:** Morné Weyers, Bianca Peterson, Josias H. Hamman, Jan H. Steenekamp

**Affiliations:** Centre of Excellence for Pharmaceutical Sciences (Pharmacen™), North-West University, Potchefstroom 2520, South Africa

**Keywords:** chitosan, fluorescein isothiocyanate dextran (FD-4), ionic gelation, microparticulate, nasal, thermosensitive gel

## Abstract

Therapeutic macromolecules (e.g., protein and peptide drugs) present bioavailability challenges via extravascular administration. The nasal route presents an alternative non-invasive route for these drugs, although low bioavailability remains challenging. Co-administration of permeation enhancers is a promising formulation approach to improve the delivery of poorly bioavailable drugs. The aim of this study was to prepare and characterize chitosan microparticulate formulations containing a macromolecular model compound (fluorescein isothiocyanate dextran 4400, FD-4) and a bioenhancer (piperine). Ionic gelation was used to produce chitosan microparticle delivery systems with two distinct microparticle sizes, differing one order of magnitude in size (±20 µm and ±200 µm). These two microparticle delivery systems were formulated into thermosensitive gels and their drug delivery performance was evaluated across ovine nasal epithelial tissues. Dissolution studies revealed a biphasic release pattern. Rheometry results demonstrated a sol-to-gel transition of the thermosensitive gel formulation at a temperature of 34 °C. The microparticles incorporating piperine showed a 1.2-fold increase in FD-4 delivery across the excised ovine nasal epithelial tissues as compared to microparticles without piperine. This study therefore contributed to advancements in ionic gelation methods for the formulation of particulate systems to enhance macromolecular nasal drug delivery.

## 1. Introduction

The nasal route of drug administration has been demonstrated to be a viable option for delivery of a broad spectrum of drugs, such as small molecular weight polar drugs, high molecular weight peptide and protein therapeutics, as well as drugs that require a rapid onset of action [1]. Since the respiratory region in the nose is characterized by a large surface area and high vascularity, this region offers optimal absorption and is therefore the main target area for nasal drug delivery [2]. The paracellular permeability of large molecules across the nasal epithelium can be enhanced by exploiting the mechanism of opening the tight junctions of the nasal epithelial cells [3].

Chitosan is a linear biopolymer consisting of glucosamine and N-acetyl glucosamine units, which can be produced by the de-acetylation of chitin [4]. Studies have demonstrated that chitosan enhances drug absorption without causing toxic effects to the nasal mucosal cells [5,6]. Illum et al. [7] investigated a chitosan-based insulin delivery system that showed promising results in the sheep model. Further studies demonstrated effective delivery of drugs in microparticle or nanoparticle chitosan formulations, which was partially attributed to chitosan’s ability to achieve muco-adhesion [8].

Chitosan forms polyelectrolyte complexes with other polymers due to the electrostatic interactions between the cationic amino groups on the C2 position and the anionic groups on the polyanions [9]. In the presence of small molecular cross-linkers, such as tripolyphosphate, chitosan forms solidified gel structures [10,11]. The process of cross-linking polymers in solution with polyelectrolytes possessing oppositely charged ions is called ionic gelation [12]. Several parameters, such as pH, stirring rate, and ambient temperature have been shown to impact the cross-linking process, which therefore determine the properties of the resulting microparticles [13]. Furthermore, previous studies have shown that it is challenging to obtain similar particle size distribution profiles as characterized by, for instance, mean particle size in replicating experiments [14,15,16,17,18].

The purpose of this study was to develop a microparticle platform for nasal delivery of macromolecules with the versatility of including bioenhancers and the potential to target a particular particle size distribution for improved nasal delivery of a macromolecular compound. The study was conducted in two phases with the first phase of this study entailing the formulation of microparticles, during which the effect of selected variables on the size of chitosan microparticles prepared by means of ionic gelation were investigated. The knowledge obtained from this phase of the study was then implemented in the second phase of the study to produce two different fluorescein isothiocyanate dextran (FD-4) containing chitosan microparticles with distinctive target sizes (i.e., ±20 µm and ±200 µm) and to evaluate these delivery systems in terms of size, size distribution, FD-4 release, and FD-4 delivery across excised ovine nasal epithelial tissues. FD-4 has been widely used in several published studies [19,20,21,22] as a macromolecular compound that is representative of a group of macromolecular drugs, therefore making it an ideal macromolecular compound for the current study.

## 2. Results and Discussion

### 2.1. Preparation of Microparticulate Formulations

Table 1 provides the d(0.9) and Span values obtained for the microparticle formulations prepared by ionic gelation with varying concentrations of chitosan (0.005–0.01 mol/L) and the cross-linking agent, tripolyphosphate (0.003–0.017 mol/L).

Figure 1 illustrates the relationship between the tripolyphosphate molar concentration (mol/L) and particle size (d(0.9)) at a fixed chitosan concentration of 0.005 mol/L. The concentration (%*w*/*v*) ratios of chitosan:tripolyphosphate are also indicated on the figure as a gradient legend to indicate the range of values represented by the color palette.

It is evident from Figure 1 that an increase in the size of the microparticles occurred with an increase in tripolyphosphate concentration. There is a direct correlation between the d(0.9) value and the tripolyphosphate concentration over the range of 0.005 mol/L and 0.017 mol/L (i.e., excluding the lowest concentration) as indicated by the coefficient of determination (R^2^) of 0.9609.

The increase in the size of the microparticles can therefore be related to the amount of cross-linking agent added to the chitosan solution, which is in line with previous findings [17]. This can be explained by the higher quantity of tripolyphosphate molecules that are available in the higher concentration solutions, which means that more chitosan molecules can be cross-linked into the microparticles. Incorporation of a higher quantity of cross-linked chitosan molecules in the microparticles led to an apparent increase in microparticle size. Changes in the tripolyphosphate concentration over the range of 0.003 to 0.017 mol/L did not have a notable influence on the Span (particle size distribution) values, which ranged only between 1.2 and 1.3.

Figure 2 illustrates the relationship between chitosan molar concentration (mol/L) and particle size (d(0.9)) at a fixed tripolyphosphate concentration of 0.006 mol/L. The concentration (%*w*/*v*) ratios of chitosan (CS):tripolyphosphate (TPP) are also indicated on the figure as a gradient legend to indicate the range of values represented by the color palette.

From Figure 2, it is clear that a U-shaped curve was obtained by plotting the microparticle size as a function of chitosan concentration. The vertex of the U-shaped curve is at 0.005 mol/L chitosan, which means that at lower as well as higher chitosan molar concentrations than 0.005 mol/L, the size of the microparticles increased. When the chitosan molar concentration was at its lowest, namely 0.002 mol/L, it resulted in the largest particle size of 156.534 µm. This corresponded to an approximate three-fold increase in the d(0.9) value from the smallest particle size of 46.763 µm. It is important to note that increases in the Span values were also observed for chitosan molar concentrations below 0.005 mol/L (Table 1) as well as at the higher concentration of 0.01 mol/L. This may be explained by aggregation or clustering of the microparticles to some extent, which led to variation in particle size distributions [23].

To test the repeatability of the ionic gelation method, microparticles were prepared for a second time using a chitosan concentration of 0.002 mol/L chitosan, which resulted in negligible differences in the particle sizes (157.113 µm and 156.534 µm). Particle size distribution (Span) results from the different tripolyphosphate and chitosan concentrations indicated that better size distribution profiles could be obtained at certain chitosan and tripolyphosphate concentration (%*w*/*v*) ratios. Based on the results obtained with the different tripolyphosphate and chitosan concentrations, the best chitosan:tripolyphosphate concentration (%*w*/*v*) ratio was found to be 1.048 (Table 1). This ratio yielded the lowest particle size distribution as indicated by the Span value.

Both chitosan and tripolyphosphate molar concentrations (chitosan: 0.005–0.01 mol/L and tripolyphosphate: 0.003–0.017 mol/L) had a marked influence on the particle size as expressed by the high standard deviations of the average d(0.9) values: 84.07 ± 44.85 for chitosan (Table 1, Figure 2) and 53.23 ± 7.27 µm for tripolyphosphate (Table 1, Figure 1). The chitosan molar concentrations had a greater effect on particle size variation, as is evident from the large standard deviation value. The chitosan molar concentrations also influenced the particle size distribution (Span value of 1.76 ± 0.65) to a greater extent as compared to that of tripolyphosphate molar concentrations (Span value of 1.26 ± 0.03).

For the small target microparticle delivery systems, the best molar concentration of chitosan was identified to be 0.005 mol/L, and tripolyphosphate molar concentration was identified to be 0.006 mol/L. These molar concentrations of the compounds yielded the smallest microparticles together with the lowest Span values. For the larger target microparticle delivery systems, the best molar concentration for chitosan was identified to be 0.002 mol/L, and tripolyphosphate molar concentration was identified to be 0.013 mol/L.

Figure 3 illustrates the effect of temperature on the size of microparticles prepared by ionic gelation, as characterized by the d(0.9) value and Span value of the particle size distribution.

Flow rate and temperature are critical parameters for chitosan microparticle preparation as is evident from previous studies. The mean particle size values obtained for chitosan microparticles related to these two variables in previous studies were in ranges of 50 to 280 µm [24,25,26]. Span values reported previously ranged between 1.4 and 5 (Cutrim et al. [27] as well as 1.6 and 1.9 (Consoli et al. [28]. Similarly, Silverio, Sakanaka, Alvim, Shirai, and Grosso [24] reported Span values between 1.7 and 2.2 for different chitosan-based ionic gelation microparticle formulations.

It is clear from Figure 3A that an increase in temperature during the ionic gelation process resulted in an increase in the size of the formed microparticles. This increase in microparticle size followed an almost linear trend with respect to increasing temperature, as evidenced by the coefficient of determination value of 0.9581. The microparticle size increased from 58.11 to 122.76 µm with an increase in temperature from 8 °C to 40 °C. The direct correlation between temperature and microparticle size may be ascribed to the influences of temperature on the protonation of chitosan. Cho et al. [29] reported on the degree of ionization of chitosan within conductivity measurements as a function of temperature. The study showed that an increase in temperature decreased the protonation of chitosan, thus suggesting a possible reduction in ionic interactions. Span values varied between 1.29 and 1.89, with a weak inverse relationship with temperature if the data point at 18 °C is omitted, as demonstrated by the coefficient of determination value of 0.8276 (Figure 3B). The narrowest particle size distribution profiles (i.e., indicated by the smallest Span value) were obtained at temperatures above 20 °C. The smaller Span values are indicative of a higher level of uniformity with respect to particle size distribution.

Figure 4 illustrates the effect of flow rate on the size and size distributions of microparticles prepared by ionic gelation.

In Figure 4, an increase in flow rate from 1 to 20 mL/min resulted in an increase in microparticle size from 41.08 to 78.33 µm. Increasing the flow rate above 20 mL/min resulted in a further pronounced increase in microparticle size of 189.07 and 216.81 µm for a flow rate of 30 mL/min and 40 mL/min, respectively. With respect to the particle size distribution (Span), low flow rates (less than 10 mL/min) resulted in the narrowest particle size distributions (i.e., smallest Span values of <1.3).

Figure 5 illustrates the effect of stirring power on the size and size distribution of microparticles prepared by ionic gelation.

It is clear form Figure 5 that stirring power influenced the size and size distribution of the microparticles prepared by ionic gelation. U-shaped curves were obtained for both microparticle size and microparticle size distribution with the vertex at 17 and 14 W/L, respectively. It can be seen that a stirring power between 9 and 12 W/L resulted in microparticle sizes greater than 80 µm. Increases in the stirring power (16–18 W/L) resulted in the smallest microparticles (~30 µm). Increasing the stirring power above 20 W/L resulted in a slight increase in the microparticle size (38 µm). A similar trend was observed for the microparticle size distributions, where a stirring power between 14 and 16 W/L yielded the narrowest particle size distribution with Span values smaller than 1.3.

Figure 6 illustrates the effect of stirring power on the size and size distribution of microparticles prepared by the different dispersing techniques of the cross-linking agent.

It can be seen from Figure 6 that, in general, the administration of the cross-linking agent via atomizing spray dispersion resulted in smaller microparticle sizes. It is also notable that the method of addition of the cross-linking agent (i.e., droplet infusion vs atomizing spray dispersion) had pronounced effects on particle size distribution (Span). Atomizing spray dispersion combined with a high stirring power (14.783 W/L) yielded the smallest microparticle sizes (d(0.9): 22.19 µm) and microparticle size distributions (Span: 0.99). A lower stirring power of 8.826 W/L in combination with the infusion dispersing technique resulted in the largest microparticles (d(0.9): 89.23 µm) with the highest microparticle size distribution (Span: 1.89). This effect can be ascribed to the forces generated during mechanical stirring on the electrostatic repulsion energy in the complexation between the negative and positive charges of tripolyphosphate and chitosan during the ionic gelation process [14]. Important to note, exceeding the electrostatic repulsion energy between positive surface charges can lead to an increase in particle aggregation [15].

The selected ionic gelation microparticles, prepared according to the parameters specified in Table 2, yielded microparticle sizes closest to that of the target microparticle sizes.

The selected microparticle formulations (Table 2) produced by ionic gelation showed microparticle sizes within the desired range, namely varying one order of magnitude from each other in size (~20 and ~200 µm). This is also in line with previous findings for microparticles obtained with ionic gelation as the overall mean microparticle size values obtained in previous studies that were reported in ranges from approximately 50 to 280 µm [24,25,26]. The Span values obtained for the two distinctive microparticle sizes were within acceptable ranges in comparison to previously reported Span values of 1.4 to 5 [27,28]. Silverio, Sakanaka, Alvim, Shirai, and Grosso [24] also demonstrated chitosan-based microparticles produced by ionic gelation with Span values between 1.67 and 2.16.

### 2.2. Validation of the Fluorescence Spectroscopy Analytical Method

Based on the regression analysis of the standard curve constructed with the fluorescence intensity values detected for a range of known FD-4 concentrations in solution, a coefficient of determination (R^2^) of 0.9983 was obtained. This indicated that the fluorescence spectroscopy analytical method had the ability to elicit fluorescence intensity values, which are directly proportional to the concentration of the analyte, FD-4, in a specific concentration range. The LOD and LOQ were determined to be 0.0058 and 0.0175 µg/mL, respectively. The sample concentrations used in the current study were well above that of the LOD and LOQ, and therefore, this analytical method could detect and accurately quantify the amount of analyte in experimental samples. The percentage relative standard deviation (%RSD) was calculated to be 3.97 and 4.25 for inter-day and intra-day precision, respectively. A %RSD ≤ 5 is deemed acceptable according to Pharmacopeia [30]. For accuracy, recovery values between 97.46% and 104.20% were obtained, thus indicating that the analytical method could accurately quantify the analyte. The fluorescence spectroscopy analytical method therefore complied with all the relevant validation criteria.

### 2.3. Preparation and Characterization of Microparticulate Delivery Systems Incorporating FD-4

Table 3 shows the experimental quantities of the FD-4 content determined for different fractions of the two formulated microparticulate delivery systems.

Calculations for small and large microparticles demonstrated an FD-4 entrapment efficiency of 59.78 and 56.24%, and an experimental net loss of 5.90 and 7.80%, respectively (Table 3). The final microparticle formulations prepared in this study demonstrated entrapment efficiency values similar to other studies where entrapment efficiencies for chitosan microparticles prepared by means of ionic gelation ranged from 27 to 73% [31,32]. An entrapment efficiency of a macromolecule (FD-4) close to 60% is considered suitable for the preparation of microparticles by ionic gelation for drug delivery purposes [33,34].

The rheological behavior of the thermosensitive gel system was expressed as a function of viscosity for each point in a temperature scan. The resulting rheological profile of the thermosensitive gel system is shown in Figure 7.

Certain rheological changes, such as the point of exponential increase in storage modulus, complex viscosity, and oscillation torque, can be used to identify the sol-to-gel transition temperature of thermosensitive gels [35,36,37,38,39]. Furthermore, since the storage modulus is an indication of the gel’s ability to store deformation energy in an elastic manner, it can be used to indicate a sol-to-gel transition point [40,41,42,43,44,45,46]. From the rheological graph (Figure 7), the sol-to-gel transition point occurred at a temperature of 34 °C, where values of storage modulus and complex viscosity increased exponentially. During the sol-to-gel transition, a change in the complex viscosity (η*) was observed for the thermosensitive gel system at temperatures between 30 and 35 °C (Figure 7). Below 34 °C, the complex viscosity of the thermosensitive gel system was very low, because it was in the sol state. The viscosity started to increase at 34 °C and continued to increase systematically with increasing temperature, indicating that the thermosensitive gel system was transitioning from the sol phase to the gel phase (i.e., a T_sol-gel_ of 34 °C). Similar results were obtained by da Silva, Cook, and Bruschi [40], where a binary thermosensitive gel system was developed, which contained poloxamer 407 (polox407) and hydroxypropyl methylcellulose (HPMC) that yielded a T_sol-gel_ of 33.8 °C when polox407:HPMC was used in a ratio of 15:2.

Since the temperature in the nasal cavity has been demonstrated to be approximately 33 to 35 °C [47,48], the aim in this study was to formulate a nasal thermosensitive gel with a T_sol-gel_ between 33 and 35 °C. The current study successfully developed a thermosensitive gel system with a T_sol-gel_ of 34 °C.

The dissolution profiles for the final microparticulate formulations for both the small microparticles (IGP20) and large microparticles (IGP200) in Krebs-Ringer bicarbonate buffer (KRB) and the thermosensitive gel system (TGS) are illustrated in Figure 8.

According to the dissolution profiles in Figure 8, all the formulations showed a rapid initial burst release of FD-4 during the first 5 min (as indicated by the relatively steep slope of the curves) and thereafter exhibited a slower release rate (as indicated by the relatively graduate slope of the curves) over the remaining dissolution period of up to 140 min. The total cumulative percentage of FD-4 released from the microparticulate formulations in Krebs-Ringer bicarbonate buffer over a period of 140 min were 94.41 ± 6.72% and 89.48 ± 6.61% for the small (IGP20) and large (IGP200) microparticles, respectively. The cumulative percentage of FD-4 released from the microparticulate formulations in thermosensitive gel systems over a period of 140 min were 94.84 ± 4.40% and 89.80 ± 3.34% for the small (IGP20) and large (IGP200) microparticles, respectively. This lower extent of release may be attributed to the relatively high viscosity of the thermosensitive gel formulations through which the FD-4 molecules must diffuse before reaching the dissolution medium.

Table 4 shows the results of kinetic analysis performed on the dissolution profiles of microparticulate formulations (with piperine) by fitting different kinetic models, namely zero-order, first-order, Higuchi, Korsmeyer–Peppas and Weibull.

The release of FD-4 from microparticulate formulations alone and in thermosensitive gel systems was modelled to determine the release kinetics and release mechanisms. Various parameters can be used to identify the mathematical model that best describes the dissolution data. The most popular parameters are the adjusted coefficient of determination (R^2^-adjusted), the akaike information criterion (AIC) and the model selection criterion (MSC). The best model is the one with the highest R^2^-adjusted, lowest AIC and highest MSC values [49]. Based on these criteria, results from Table 4 indicated that both microparticulate formulations (with piperine) in Krebs-Ringer bicarbonate buffer and the thermosensitive gel system followed the Korsmeyer–Peppas model of release (Equation (16)). Values of 0.126 and 0.208 were obtained for the Korsmeyer–Peppas shape parameter (*n*) for the IGP20 and IGP200 microparticulate formulations (with piperine) in Krebs-Ringer bicarbonate, respectively. Values less than 0.5 indicate that the release of FD-4 followed a Fickian mechanism of diffusion, which means that the rate of release was independent of the FD-4 concentration in the formulations. More specifically, Fickian diffusion occurs when the polymer relaxation time (*t_r_*) is much greater than the characteristic solvent diffusion time (*t_d_*) [50]. A Fickian profile is characterized by an initial linear release as a function of time (t), followed by a second phase with a linear release as a function of t^1/2^. This release behavior was characteristic of the FD-4 release, as can be seen in Figure 8. For both the Krebs-Ringer bicarbonate buffer and the thermosensitive gel system, the cumulative percentage FD-4 released was slightly higher for IGP20 than for IGP200.

However, the release mechanism seemed to be unaffected by microparticle size. A previous study on formulations of hydrocortisone butyrate (HB)-loaded poly (D,L-lactic-co-glycolic acid) nanoparticles (PLGA NP) also demonstrated a biphasic release pattern with an initial burst release phase (steep slope) followed by a sustained release phase (gradual slope) [51]. However, when the PLGA NP were suspended in thermosensitive gel in that study, zero-order release kinetics was observed. This was not the case in the current study, since microparticulate formulations (with piperine) in both Krebs-Ringer bicarbonate buffer and the thermosensitive gel system followed the Korsmeyer–Peppas model of release, which indicates modified release.

### 2.4. Ex Vivo Permeation Studies

The cumulative percentage transport of FD-4 from microparticulate formulations (with and without piperine) in Krebs-Ringer bicarbonate buffer and the thermosensitive gel system across excised ovine nasal tissue are illustrated in Figure 9.

The cumulative percentage permeation of FD-4 achieved from the microparticulate formulations with and without piperine were higher than that of the control (FD-4 alone, 0.296 ± 0.170%). The cumulative percentage transport of FD-4 achieved from the microparticulate formulations with piperine in the thermosensitive gel system were 0.295 ± 0.145% and 0.347 ± 0.086% for IGP20 (IGP20.Piperine_TGS) and IGP200 (IGP200.Piperine_TGS), respectively (Figure 9). The IGP200 microparticulate formulation with piperine in the thermosensitive gel system (IGP200.Piperine_TGS) produced a higher cumulative percentage transport than that of the control (FD-4 alone, 0.296 ± 0.170%) with piperine (Control.Piperine), whereas IGP20 in piperine (IGP20.Piperine_TGS) yielded similar cumulative percentage transport of FD-4 to that of the control (FD-4 alone, 0.296 ± 0.170%) with piperine (Control.Piperine).

The average permeability coefficient (P_app_) values of FD-4 microparticulate formulations with and without piperine in both Krebs-Ringer bicarbonate (KRB) buffer and the thermosensitive gel system (TGS) across excised ovine nasal tissue are illustrated in Figure 10.

Microparticulate formulations in Krebs-Ringer bicarbonate buffer and the thermosensitive gel system yielded apparent permeability coefficient (P_app_) values for FD-4 between 1.66 × 10^−7^ and 2.81 × 10^−7^ cm/s across excised ovine nasal tissue (Figure 10). Although there were no statistically significant differences for FD-4 transport between the control and the experimental solutions (Krebs-Ringer bicarbonate buffer and thermosensitive gel system), the piperine incorporated microparticulate formulations demonstrated a slight increase in overall average apparent permeability in comparison to the microparticulate formulations not containing piperine. The lowest and highest percentage difference in apparent permeability between microparticulate formulations were 10.12% (without piperine) and 29.57% (with piperine), respectively. Previously published studies indicated that piperine enhances polarized paracellular transport by opening tight junctions [52,53,54].

In comparison, an in situ gel of venlafaxine displayed higher permeability (P_app_ ≈ 5.9 × 10^−6^ cm/s) in an ex vivo study with sheep nasal mucosa [47]. The lower permeability values observed in the current study may be attributed to the larger size of the model compound, FD-4 (4400 g/mol), in comparison to that of venlafaxine (313.9 g/mol) [47].

Average TEER values decreased from 68.1 ± 2.3 to 54.1 ± 3.0 Ω·cm^2^ in the presence of formulations without piperine, whereas the average TEER values decreased from 63.3 ± 3.7 to 44.7 ± 2.2 in the presence of formulations with piperine. A previous study by Gerber, Steyn, Kotzé, Svitina, Weldon, and Hamman [54] obtained similar results, where the decrease in TEER values was ascribed to potential opening of tight junctions. The lower TEER values observed for formulations containing piperine in the current study may therefore be indicative of tight junction modulation by piperine, which may have contributed to the slightly higher transport of FD-4.

TEER values previously observed for human nasal mucosa were 75 to 100 Ω·cm^2^, whereas rabbit, bovine, and porcine nasal mucosa yielded TEER values of 52, 40–200, and 68–74 Ω·cm^2^, respectively [55,56]. Contrarily, another study showed low TEER values ranging from 21 to 30 for porcine nasal mucosa [56]. According to Srinivasan et al. [57] factors, such as temperature, medium formulation, and passage number of cells, can contribute to variations in TEER values [57]. The TEER values obtained in this study were therefore slightly lower than that observed for human nasal mucosa [55]. Nonetheless, the TEER values obtained in this study are in range with that of other studies conducted on animal tissue models mentioned previously [55,56], which confirms that the integrity of the sheep nasal mucosal tissue was maintained for the duration of the study.

## 3. Conclusions

The current study aimed to determine the effect of selected factors on the size of chitosan microparticles prepared by means of ionic gelation. Larger microparticles were produced by using the infusing droplet dispersing technique with high chitosan and tripolyphosphate molar concentrations, temperature, and flow rate, together with a slow stirring speed. In contrast, small microparticles were obtained by using the atomizing spray dispersing technique with low chitosan and tripolyphosphate molar concentrations, temperature, and flowrate, while applying a relatively fast stirring rate. The optimal parameters were subsequently used to successfully produce two different FD-4 containing chitosan microparticles with distinctive target sizes (i.e., ±20 µm and ±200 µm). The microparticulate dosage forms prepared by means of ionic gelation containing FD-4 and piperine have demonstrated the ability to deliver macromolecular compounds across excised nasal epithelial tissues and may provide a promising alternative for non-invasive macromolecular compound delivery.

## 4. Materials and Methods

### 4.1. Materials

FD-4 (MW = 4400 g/mol, CAS: 60842-46-8), piperine (CAS: 94-62-2), Pluronic^®^ F-127 (CAS: 9003-11-6), hydroxypropyl-methyl cellulose (HPMC, CAS: 9004-65-3), tripolyphosphate (CAS: 7758-29-4), and Krebs-Ringer bicarbonate buffer (CAS: 144-55-8) were purchased from Sigma-Aldrich (Johannesburg, South Africa). Chitosan (CAS: 9012-76-4) was sourced from Warren Chem Specialities (Johannesburg, South Africa). Purified water was prepared with a Milli Q water purification system (Millipore, South Africa). Nasal epithelial tissues were excised from sheep at a local abattoir (Potchefstroom, South Africa) after slaughtering of the sheep for meat production purposes. The use of excised sheep nasal tissues for research purposes was approved by the North-West University animal ethics committee (ethics application approval no. NWU-00285-17-A5).

### 4.2. Formulation (Phase 1)

#### 4.2.1. Factors That Determine Microparticle Size during Ionic Gelation

Microparticles were prepared by ionic cross-linking of chitosan with tripolyphosphate according to a previously described procedure [11]. In principle, the method entailed the addition of a tripolyphosphate solution to a chitosan solution under stirring. The effect of selected factors on the size of chitosan microparticles that were produced with the ionic gelation method was systematically investigated in order to determine the specific manufacturing parameters that would be needed to prepare the two-target chitosan microparticle systems incorporating FD-4 (MW = 4400 Da) as a model macromolecular compound. These factors included concentration of chitosan and tripolyphosphate, temperature, flow rate during addition of the cross-linking agent, the kinetic power during stirring (stirring power), and the technique of adding the cross-linking solution to the chitosan solution. Each of these factors are discussed in detail below.

The effect of the concentrations of chitosan (0.005–0.01 mol/L) and tripolyphosphate (0.003–0.017 mol/L) on the size of the chitosan microparticles obtained during ionic gelation was investigated (Table 5). To maintain consistency in the resultant particle size distribution, the initial tests were performed in accordance with the current literature available on ionic gelation methodology [58,59,60,61]. The following parameters were used during the ionic gelation method: rotor speed of 500 rpm, flow rate of 5 mL/min, and a temperature of 18 °C. Firstly, variable volumes of tripolyphosphate solutions were administered to determine the optimum tripolyphosphate concentration (Table 5), as characterized by the particle size value (d(0.9)) and the width of the distribution (Span). The d(0.9) value represents a standard percentile reading from the particle size analysis, which is the particle size value below which 90% of the sample falls with regard to particle size [62]. Span is calculated with the following equation: (1)$$\mathrm{Span}=\frac{d\left(0.9\right)-d\left(0.1\right)}{d\left(0.5\right)}$$

The values of d(0.9), d(0.1) and d(0.5) represent the particle size values below which 90%, 10% and 50% of the sample falls with regard to particle size, respectively. The d(0.5) value is also known as the median. A larger Span value indicates a wider particle size distribution whereas a smaller Span value indicates a narrower particle size distribution [62].

Subsequently, the optimized tripolyphosphate concentration was administered in the ionic gelation method with varying chitosan concentrations (Table 5).

The concentration ratios (%*w*/*v*; chitosan:tripolyphosphate) were calculated with the following equation:(2)$$\mathrm{Concentration}\;\mathrm{ratio}\;(\mathrm{chitosan}:\mathrm{tripolyphosphate})=\frac{\mathrm{chitosan}\;\mathrm{concentration}\;(\%w/v)}{\mathrm{tripolyphosphate}\;\mathrm{concentration}\;(\%w/v)}$$

Based on the particle size (d(0.9)) and the particle size distribution (Span) values obtained for the microparticles as prepared with the parameters indicated in Table 5, specific tripolyphosphate and chitosan concentrations were selected for further microparticle production as used in subsequent experiments.

The temperature (°C) is specified as the chitosan solution temperature at the commencement of the addition of the tripolyphosphate solution. The temperature of the chitosan solution was varied between 8 and 40 °C in the current study. In this study, flow rate refers to the rate of addition of the tripolyphosphate solution into the chitosan solution during microparticle preparation by means of ionic gelation. The flow rate was varied between 1 and 40 mL/min in the current study. The optimum flow rates and temperatures, together with the other parameters, were selected based on the particle sizes obtained to produce the target microparticulate formulations.

The calculated stirring power refers to the force applied by the impeller onto the liquid medium during the ionic gelation process. Stirring power is a newly proposed variable that takes stirring speed, impeller design dimensions, external mechanical factors, and changes in solution viscosity into account. While preparing the microparticles by means of the ionic gelation process, stirring was performed with an electronic Heidolph RZR adjustable torque rotor driving system, which constituted the stirring apparatus when equipped with an impeller. For this study, a modified stirring rod with a custom impeller (3 blades, 33.5 mm in diameter at a 35° angle) was inserted in the torque rotor driving system. From the impeller diameter and adjustable rotation rate settings, the stirring speed was calculated by using Equation (3).
(3)$$\mathrm{Stirring}\;\mathrm{speed}\;(m/s)=\frac{\mathrm{RPM}\;\times \;\pi \;\times \;Ø}{60}$$

The rotor driving system’s kinetic power output (KPO) (W) during stirring was calculated by means of Equation (4). The electric potential (V) and current (A) were measured by connecting a Fluke 289 true RMS multi-meter apparatus to the rotor driving system and taking readings when stirring the solutions during ionic gelation.
(4)KPO (W)=current (A) × electric potential (V)

By subtracting the kinetic power output (KPO) of a running stirring apparatus in the presence of a solution from the kinetic power output (KPO) of the same running stirring apparatus without solution present, the total mechanical power (TMP) exerted on the solution during stirring at a specific stirring speed can be calculated. The TMP was calculated using Equation (5).
(5)TMP (W)= KPO (W) with solution − KPO (W) without solution
where TMP is the total mechanical power, and KPO is the kinetic power output.

The stirring power (W/L) was subsequently calculated using the following equation:(6)$$\mathrm{Stirring}\;\mathrm{power}\;\left(W/L\right)=\left[\frac{\mathrm{total}\;\mathrm{mechanical}\;\mathrm{power}\;(W)}{\mathrm{solution}\;\mathrm{volume}\;\left(\mathrm{mL}\right)}\right]\times \;1000$$

Besides the addition of tripolyphosphate solution to the chitosan solution through continuous flow by means of peristaltic pumping (i.e., infusion droplet technique), an atomizing spray dispersing technique using an Aircraft pneumatic system with manually adjustable atomizing pressure (i.e., spray dispersion technique) was also applied. The effect of the technique of cross-linking solution addition on the size of the microparticles formed during ionic gelation was therefore investigated with this experiment. With administration of the tripolyphosphate solution by means of the atomizing spray dispersing technique, the distance of the atomizer spray nozzle (professional airbrush kit from Aircraft pneumatic systems) was kept constant at 7.5 cm to the surface of the chitosan solution during the ionic gelation process. For comparison, two fixed continuous stirring powers of 8.83 W/L and 14.78 W/L were applied during the ionic gelation method for both the spray dispersion and infusion droplet techniques, while keeping the solution at 18 °C.

#### 4.2.2. Validation of the Fluorescence Spectroscopy Analytical Method

The fluorescence spectroscopic analytical method that was used to measure FD-4 concentrations in the samples was validated in terms of linearity, limit of detection (LOD), limit of quantification (LOQ), and precision.

A stock solution of 500 µg/mL of FD-4 was prepared in Krebs-Ringer bicarbonate buffer. A serial dilution of eight solutions with different concentrations (0.98, 1.95, 3.91, 7.81, 15.63, 31.25, 62.5, and 125 µg/mL) were prepared from the stock solution by transferring aliquots to tubes containing Krebs-Ringer bicarbonate buffer. The fluorescence intensity of each concentration was measured in triplicate. A calibration curve was constructed from the spectrofluorimetric data, after which linear regression analysis was conducted. These values were also used to calculate the LOD and LOQ using Equations (7) and (8).
(7)$$\mathrm{LOD}=\frac{3.3\;\sigma }{S}$$
(8)$$\mathrm{LOQ}=\frac{10\;\sigma }{S}$$
where σ is the standard deviation of the response, and S is the slope of the calibration curve [63].

To determine the precision and accuracy of the analytical method, solutions of FD-4 with three different concentrations (12.5, 62.5, and 125 µg/mL) were prepared in Krebs-Ringer bicarbonate buffer and the relative fluorescence intensity of each solution was measured in triplicate. The percentage recovery (i.e., accuracy) was calculated using Equation (9):(9)$$\mathrm{Percentage}\;\mathrm{recovery}=\frac{\left[\mathrm{actual}\;\mathrm{concentration}\right]}{\left[\mathrm{theoretical}\;\mathrm{concentration}\right]}\;\times \;100$$

The percentage relative standard deviation (%RSD) was calculated to confirm the precision of the analytical method with Equation (10):(10)$$\%\mathrm{RSD}=\;\frac{\mathrm{SD}}{\mathrm{mean}}\;\times \;100$$
where %RSD is the percentage relative standard deviation, SD is the standard deviation of the fluorescence intensity values of sample replicates, and mean is the average fluorescence intensity of sample replicates.

#### 4.2.3. Preparation of the Target Microparticle Delivery Systems Incorporating FD-4

From the results obtained during the first phase of the study, the appropriate preparation conditions to produce the target microparticle sizes (i.e., ±20 µm and ±200 µm) were selected.

The atomizing spray addition technique was used to produce the smaller target microparticles, while using concentrations of 0.005 and 0.006 mol/L chitosan and tripolyphosphate, respectively. A stirring speed with a stirring power of 17.17 W/L was applied to the chitosan solution with addition of the cross-linking agent (tripolyphosphate). The atomizing spray system was set at 0.6 Barr of pressure and a flow rate of 2.5 mL/min, while the solution temperature was set at 8 °C. To obtain the larger microparticles, the infusion droplet dispersing technique was used with a flow rate of 25 mL/min. Chitosan and tripolyphosphate molar concentrations were 0.02 and 0.013 mol/L, respectively. A stirring speed with a stirring power of 8.8 W/L was applied to the chitosan solution with addition of the cross-linking agent, while the solution temperature was set at 40 °C. FD-4 (0.05%) and piperine (0.007%) were added into the chitosan solution and stirred before addition of the cross-linking agent. FD-4 has been extensively used in ex vivo transport studies as a model compound for macromolecular drug permeation, mainly to investigate membrane permeation enhancement via the paracellular pathway by means of tight junction modulation [64,65,66]. The microparticles that were produced by ionic gelation as described above, which contained the FD-4 and piperine, were then lyophilized for at least 72 h using a Virtis benchtop freeze dryer (United Scientific, Gauteng, South Africa). The dry materials were then transferred into air-tight glass containers and stored in a desiccator until required for dissolution and transport studies. Microparticles incorporating FD-4 were also prepared without piperine to examine the drug delivery capabilities of microparticles with and without a bioenhancer across excised ovine nasal epithelial tissues.

A sample (40 mg) from each microparticle drug delivery system was transferred to a volumetric flask and made up to volume (10 mL) with purified water and gently agitated for 30 s to wash off and dissolve any FD-4 molecules that may be present on the surface of the microparticle delivery systems. The suspension was filtered through a 0.45 µm syringe filter and the filtrate represented the wash-out external surface FD-4 content. The microparticles that remained after the filtration process (representing microparticles without external surface FD-4) were collected and transferred to a volumetric flask and made up to volume (10 mL) with purified water. The microparticle dispersion was placed into an ultrasonic bath for 25 min and then magnetically stirred for a period of 3 h. The microparticle dispersion was then filtered through a 0.45 µm syringe filter and the filtrate represented the releasable FD-4 content of the microparticle delivery systems. The microparticles obtained from the latter filtration process were collected, physically crushed, and transferred to a volumetric flask (10 mL), which was made up to volume with purified water and placed into an ultrasonic bath for 25 min and then magnetically stirred for a period of 3 h. The dispersion was then filtered through a 0.45 µm syringe filter, and the filtrate represented the non-releasable FD-4 quantity that is bound to the polymeric part of the microparticles. The FD-4 concentration of each filtrate was quantified by means of fluorescence spectroscopy using a Spectramax Paradigm multimode detection platform plate reader. The fluorescence analysis was conducted at an excitation and emission wavelength of 485 nm and 525 nm, respectively [64]. The quantity of the FD-4 in the microparticle delivery systems was calculated from the calibration curve. Equation (11) was used to calculate the FD-4 entrapment efficiency of the two microparticle delivery systems.
(11)$$\%\;\mathrm{Entrapment}\;\mathrm{efficiency}=\frac{\mathrm{experimental}\;\mathrm{value}\;\mathrm{of}\;\mathrm{FD-4}\;\mathrm{content}}{\mathrm{theoretical}\;\mathrm{value}\;\mathrm{of}\;\mathrm{FD-4}\;\mathrm{content}}\;\times \;100$$

The net loss of FD-4 content was then calculated using Equation (12):(12)$$\mathrm{Net}\;\mathrm{loss}=100\;-\left[\frac{\mathrm{experimetal}\;\mathrm{value}\;\mathrm{of}\;\mathrm{FD-4}\;\mathrm{content}}{\mathrm{theoretical}\;\mathrm{value}\;\mathrm{of}\;\mathrm{FD-4}\;\mathrm{content}}\;\times \;100\right]$$

The basic thermosensitive gel formulation, which consisted of 5% *w*/*v* hydroxypropyl methyl cellulose (HPMC) and 20% *w*/*v* Pluronic^®^ F127 (Poloxamer 407), was prepared according to the methods described by Pund et al. [47]. A cooling bath was used during the addition of Pluronic^®^ F127 to distilled water (4 °C) with continuous stirring for 5 min. Afterward, it was set aside for 8 h to reach thermal equilibrium at ambient temperature (20 °C). The resultant gel base formulation was stored overnight at 4 °C [47,67].

The nasal thermosensitive gel system was obtained by mixing the thermosensitive gel formulation and Krebs-Ringer bicarbonate buffer in a 1:1 ratio. From its freeze-dried powder form, each of the two distinctly sized microparticulate formulations was mixed with the nasal thermosensitive gel system to form the final thermosensitive nasal gel formulations that were characterized and evaluated for FD-4 (i.e., macromolecular compound) delivery in ex vivo permeation studies.

Rheological behavior of the thermosensitive gel was characterized with an ARES-G2 rheometer (TA Instruments, New Castle, DE, USA) using oscillatory rheometry, which is a non-destructive technique that can be employed for investigating structure changes in materials. Characterization was performed with an oscillation temperature ramp viscosity protocol to obtain the rheological behavior of both the sol and gel state. Execution of the viscosity experiments was performed with 1 °C incremental temperature increases with active acclimatization, starting from 25 °C and ending at 40 °C. The frequency, strain, and sampling rate were held constant at 1.67 Hz, 0.5%, and 1 pts/s, respectively. Oscillatory shear rheology is commonly used to determine this gelation-point [35,40,41]. During the sol-to-gel transition, the storage modulus (G’) increases when plotted against temperature [35,36,37]. Since the storage modulus is an indication of the gel’s ability to store deformation energy in an elastic manner, it can therefore be used for indicating a sol-to-gel transition point (i.e., the point of gelation) [40,41,42,43,44,45,46]. The complex viscosity and oscillation torque indicate the change in oscillatory shear stress within the gel formulation, which also assists in identification of a sol-to-gel transition point [40,41,42,43,44,45,46].

The gelation characteristics of the thermosensitive gel systems are presented as thermosensitive shear properties and the storage modulus (G’), complex viscosity (η*), and oscillation torque (M) plotted as a function of temperature (T).

FD-4 release from both the microparticulate formulations and the nasal thermosensitive gel formulations was measured using a Distek 2500 dissolution system, coupled with an Evolution 4300 autosampler and dual syringe pump apparatus (121 North Center Drive, North Brunswick, NJ 08902, USA). The dissolution medium consisted of freshly prepared Krebs-Ringer bicarbonate buffer. For both dissolution studies, the solution medium was added to Distek dissolution vessels with a total solution volume of 400 mL, while being stirred continuously at 50 rpm at 37 ± 0.5 °C. Samples (1 mL), filtered through a 0.45 µm syringe filter, were withdrawn from each dissolution vessel at time intervals of 1, 5, 10, 20, 40, 60, 80, 100, 120, and 140 min, which was immediately replaced with 1 mL of Krebs-Ringer bicarbonate solution at each withdrawal point.

The samples were analyzed by means of the validated fluorescence spectroscopy analytical method. The fluorescence data were corrected for dilution, then processed and finally illustrated as the cumulative percentage FD-4 released as a function of time [68].

The dissolution data were subsequently analyzed with DDSolver^®^ (an Excel Add-In software program, Nanjing, China) to determine the in vitro drug release kinetics of the microparticulate formulations [49]. The FD-4 release profiles of the microparticles were investigated by using various mathematical models, namely zero order (Equation (13)), first order (Equation (14)), Higuchi (Equation (15)), Korsmeyer–Peppas (Equation (16)) and Weibull (Equation (17)).
(13)$${F=k}_{0}\;\cdot \;t$$
(14)$$F=100\;\cdot \;\left({1-\;e}^{k_{1}\cdot t}\right)$$
(15)$${F=k}_{H}{\;\cdot \;t}^{0.5}$$
(16)$${F=k}_{\mathrm{KP}}\;\cdot \;t^{n}$$
(17)$$F=100\;\cdot \;\left\{{1-\;e}^{\frac{t-{\mathrm{Tt}}^{\beta }}{\alpha }}\right\}$$
where F is the fraction of drug released at time t, k is the release constant for each model, n is the diffusional exponent indicating the drug-release mechanism, α is the scale parameter, which defines the time scale of the process, β is the shape parameter, which characterizes the shape of the curve, and Tt is the location parameter, which represents the lag time before the onset of the dissolution or release process [69].

### 4.3. Ex Vivo Permeation Studies (Phase 2)

Nasal mucosa was excised from the noses of sheep, which was collected directly after slaughter of the sheep (for meat production purposes) at the local abattoir in Potchefstroom, South Africa (ethics application approval no. NWU-00285-17-A5). The upper jaw was removed from the sheep skull by making a longitudinal incision through the lateral wall (behind the *incisura nasoincisiva*), after which it was rinsed, submerged, and transported to the laboratory in fresh, ice-cold Krebs-Ringer bicarbonate buffer. The frontal parts of the nasal conchae were removed by excision and the mucosal epithelial tissue was stripped from the lateral cartilage by means of blunt dissection [70,71].

All the ex vivo permeation studies were conducted in a Sweetana–Grass diffusion chamber apparatus (Harvard NaviCyte apparatus, Warner Instruments, Palo Alto, California, USA), which was connected to a heating block (34 °C) and carbogen supply (95% O_2_:5% CO_2_). Strips of approximately 1 cm (width) were cut from the isolated ovine nasal epithelial tissues and these were mounted between the half-cells of the diffusion chambers within 30 min after removing it from the sheep head.

Before commencement of the permeation studies, both half cells of the chamber were incubated with 7 mL pre-heated Krebs-Ringer bicarbonate buffer (pH 6.4) for 15 min. For measurement of FD-4 permeation across the excised sheep nasal epithelial tissues from the control group, 7 mL aliquots of FD-4 dissolved in Krebs-Ringer bicarbonate buffer (500 µg/mL FD-4) were added to the donor chambers. For measurement of FD-4 permeation across the excised sheep nasal epithelial tissues from the experimental microparticulate formulations, 7 mL aliquots of microparticles suspended in Krebs-Ringer bicarbonate buffer (equivalent to 500 µg/mL FD-4) were added to the donor chamber. For measurement of FD-4 permeation across the excised sheep nasal epithelial tissues from the nasal thermosensitive gel systems, 7 mL aliquots of the gel formulations were added to the donor chamber. In all these permeation studies, the acceptor chamber was filled with 7 mL Krebs-Ringer bicarbonate buffer (pH 6.4), which was kept at a constant temperature of 34 °C while being stirred continuously. Samples of 180 µL were withdrawn from the basolateral side at 20 min intervals and replaced with equal volumes of pre-heated Krebs-Ringer bicarbonate buffer, over a total time period of 2 h [47,48,68]. The samples were analyzed by means of the validated fluorescence spectroscopy analytical method, and the resulting data were analyzed to calculate the cumulative percentage FD-4 transported across excised ovine nasal epithelial tissue.

The FD-4 concentration in each sample was corrected for dilution before calculating the cumulative percentage transport of FD-4 across the excised ovine nasal epithelial tissues. Subsequently, the apparent permeability coefficient (P_app_) values were calculated using Equation (18).
(18)$$P_{\mathrm{app}}=\frac{\mathrm{dQ}}{\mathrm{dt}}\left(\frac{1}{A.60.C_{0}}\right)$$
where P_app_ is the apparent permeability coefficient (cm·s^−1^), $\frac{\mathrm{dQ}}{\mathrm{dt}}$ is the permeability rate (amount permeated/minute), A is the diffusion area of the membrane (cm^2^), and C_0_ is the initial concentration of FD-4 incubated on the membrane [68,72].

Barrier function of biological membranes is easily quantifiable by measuring the trans-epithelial electrical resistance (TEER) across the epithelial mucosal membrane [55]. In this study, TEER was measured every 20 min, for the duration of the study, to determine whether any changes in tight junctions or membrane integrity occurred. TEER measurements were performed with a EC-825A epithelial voltage clamp (Serial nr 211, Warner Instruments^®^, Hamden, CT, USA).

### 4.4. Data Analysis

Experimental data obtained from the particle size and size distribution test during the investigation of selected factors during ionic gelation were visualized with the ggplot2 software package using the RStudio program (R Core Team, 2020) (Wickham et al. [73]). Results obtained from ex vivo permeation studies were analyzed (corrected for dilution and P_app_ calculated using Equation (18)) and visualized in Excel (Microsoft Office 365).

To determine if there were any statistically significant differences between the P_app_ values of the control groups (FD-4 with and without piperine) and experimental groups (microparticulate formulations), a one-way analysis of variance (ANOVA) was performed, while a non-parametric analysis was conducted by means of the Tukey HSD and Kruskal–Wallis post-hoc tests [74]. Statistically significant differences were accepted when *p* < 0.05.

## Figures and Tables

**Figure 1 gels-08-00686-f001:**
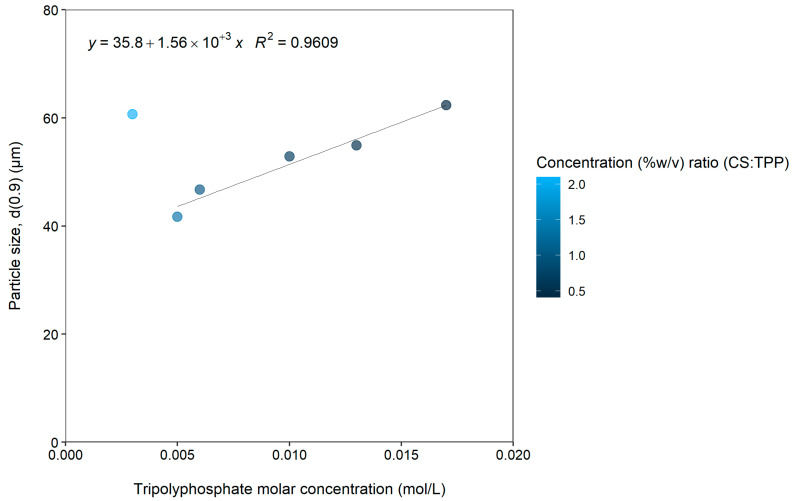
The particle size (d(0.9)) values plotted as a function of tripolyphosphate molar concentration at a fixed chitosan concentration of 0.005 mol/L. The chitosan (CS):tripolyphosphate (TPP) concentration (%*w*/*v*) ratios are also indicated by color intensity of the markers.

**Figure 2 gels-08-00686-f002:**
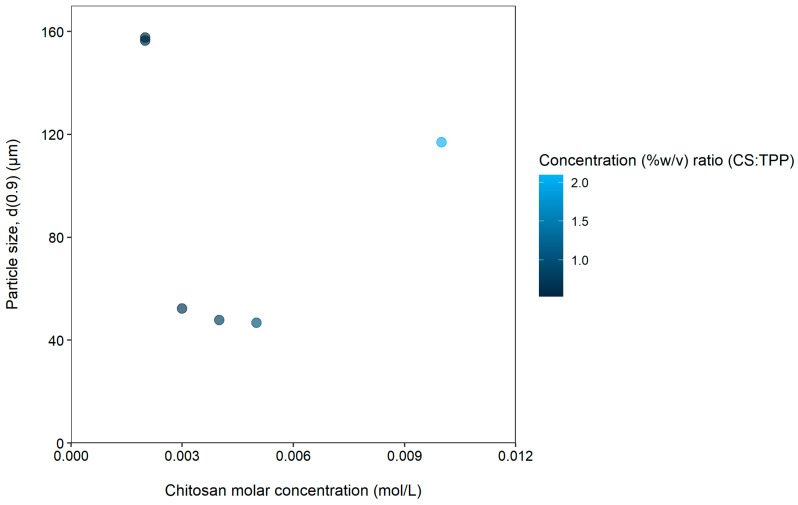
The particle size (d(0.9)) plotted as a function of chitosan molar concentration at a fixed tripolyphosphate concentration of 0.006 mol/L. The chitosan (CS):tripolyphosphate (TPP) concentration (%*w*/*v*) ratios are also indicated by color intensity of the markers.

**Figure 3 gels-08-00686-f003:**
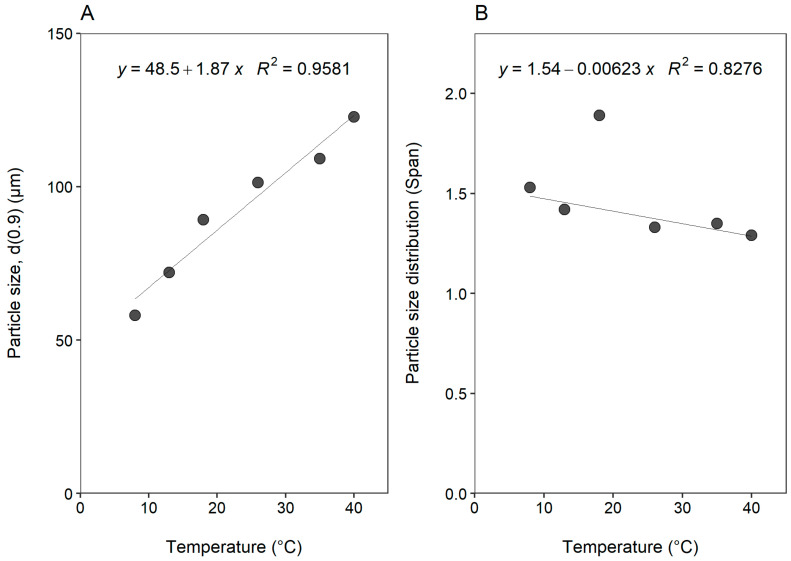
The effect of temperature on (**A**) particle size (d(0.9)) and (**B**) particle size distribution (Span).

**Figure 4 gels-08-00686-f004:**
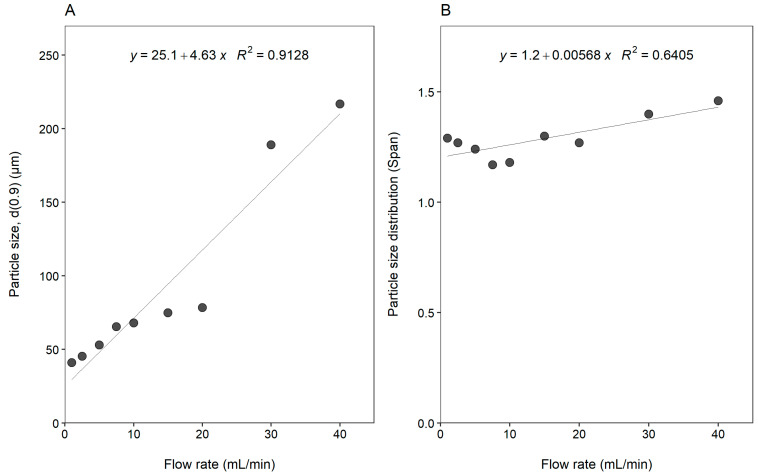
The effect of flow rate (mL/min) on (**A**) particle size (d(0.9)) and (**B**) particle size distribution (Span).

**Figure 5 gels-08-00686-f005:**
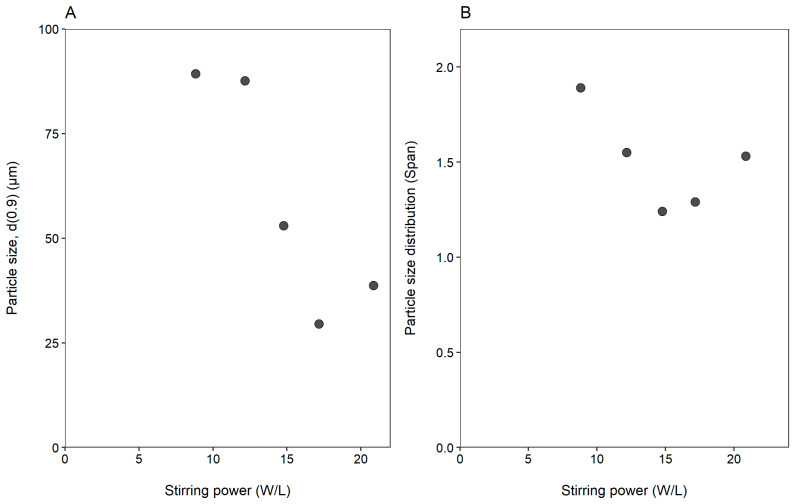
The effect of stirring power (W/L) on the (**A**) particle size (d(0.9)) and (**B**) particle size distribution (Span).

**Figure 6 gels-08-00686-f006:**
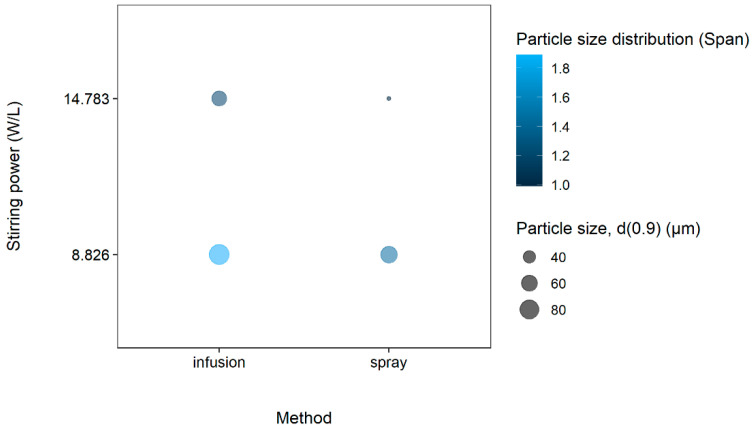
The effect of stirring power (W/L) and method of adding the cross-linking agent on the microparticle size as (d(0.9)) and microparticle size distribution (Span).

**Figure 7 gels-08-00686-f007:**
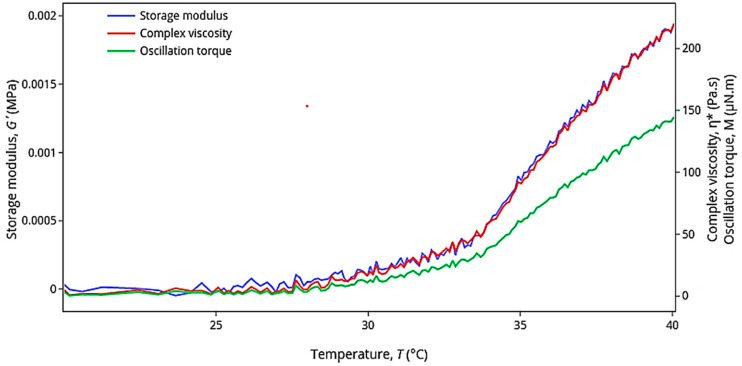
Rheological graph of the thermosensitive gel system expressing the storage modulus (G’), complex viscosity (η*), and oscillation torque (M) as a function of temperature (T).

**Figure 8 gels-08-00686-f008:**
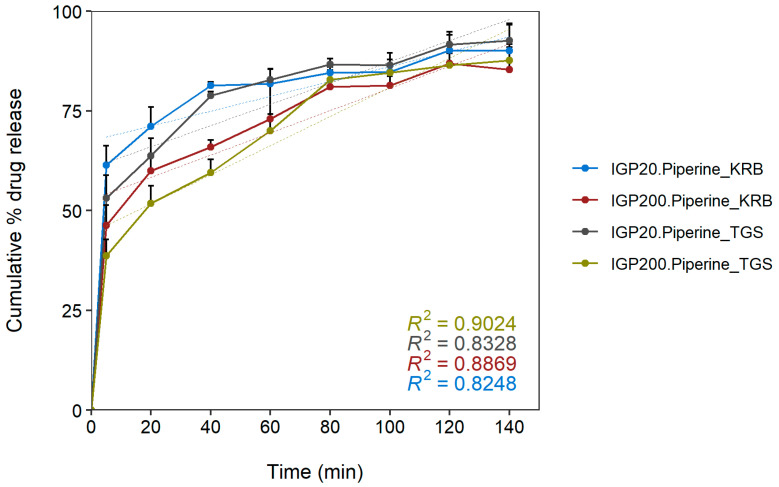
Dissolution profiles for the microparticle formulations (IGP20 and IGP200) and thermosensitive gel in Krebs-Ringer bicarbonate buffer. Data are expressed as the mean cumulative percentage FD-4 release ± standard deviation; n = 6. IGP20: formulations aimed for microparticle size of ±20 µm, IGP200: formulations aimed for microparticle size of ±200 µm, KRB: Krebs-Ringer bicarbonate buffer, TGS: thermosensitive gel system.

**Figure 9 gels-08-00686-f009:**
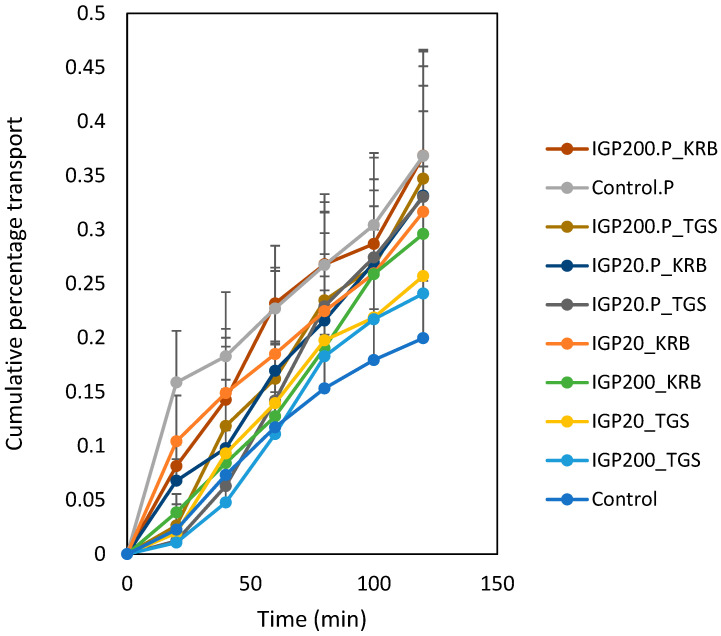
Apical-to-basolateral cumulative permeation of FD-4 alone (control) and from small microparticles (IGP20) and large microparticles (IGP200) with and without piperine (P) in Krebs-Ringer bicarbonate buffer (KRB) and the thermosensitive gel system (TGS) across excised ovine nasal tissue. Data are expressed as the mean cumulative percentage permeation ± standard deviation; n = 6.

**Figure 10 gels-08-00686-f010:**
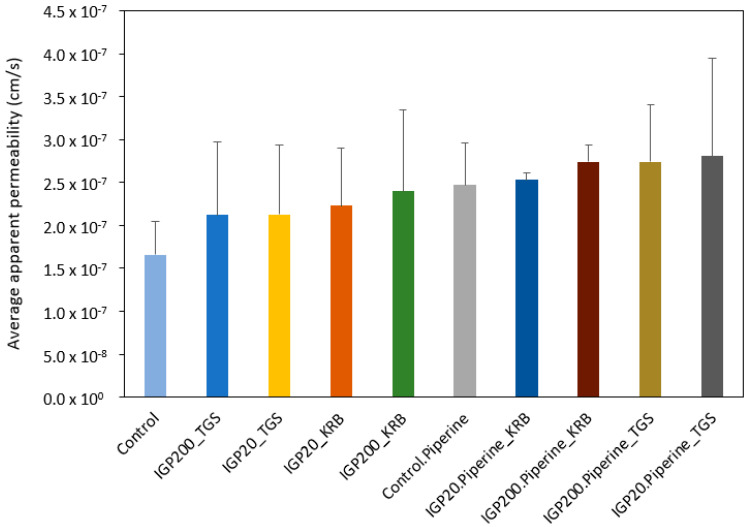
Average permeability coefficient values of FD-4 microparticulate formulations, with and without piperine, in Krebs-Ringer bicarbonate buffer (KRB) and nasal thermosensitive gel systems (TGS) across excised ovine nasal tissue. Data are expressed as the mean apparent permeability coefficient (P_app_) values ± standard deviation; no statistically significant differences were observed, n = 6. IGP20: small microparticles (±20 µm), IGP200: large microparticles (±200 µm).

**Table 1 gels-08-00686-t001:** Particle size (d(0.9)) and particle size distribution (Span) values obtained for microparticles prepared with various molar concentrations and concentration (%*w*/*v*) ratios of the tripolyphosphate and chitosan solutions.

Formulation	CS Molar Concentration (mol/L)	TPP Molar Concentration (mol/L)	Concentration (%*w*/*v*) Ratio (CS:TPP) *	Span	d(0.9) (µm)
1	0.005	0.017	0.409	1.28	62.385
2		0.013	0.524	1.24	54.913
3		0.01	0.699	1.25	52.906
4		0.006	1.048	1.22	46.763
5		0.005	1.395	1.3	41.707
6		0.003	2.097	1.25	60.709
7	0.002	0.006	0.524	1.8	157.692
8	0.002		0.524	3.06	156.534
9	0.003		0.699	1.24	52.286
10	0.004		0.839	1.26	47.769
11	0.005		1.048	1.22	46.763
12	0.01		2.097	2	117.014

* CS = chitosan; TPP = tripolyphosphate.

**Table 2 gels-08-00686-t002:** Particle size (d(0.9)) and size distribution (Span) values for two target microparticles obtained from ionic gelation with specific parameters.

Parameter	Small Microparticles	Large Microparticles
d(0.9) (µm)	22.194	190.218
Span	0.986	2.099
Chitosan molar concentration (mol/L)	0.005	0.02
Tripolyphosphate molar concentration (mol/L)	0.006	0.013
Dispersing technique	Atomizing spray	Infusion droplet
Flow rate (mL/min)	2.5	25
Stirring power (W/L)	17.17	8.8
Temperature (°C)	8	40

**Table 3 gels-08-00686-t003:** FD-4 content in different fractions of the microparticle delivery systems.

Fraction	FD-4 Concentration (mol/L)	
	Small Microparticle Delivery System	Large Microparticle Delivery System
Fraction on external surface of microparticles	0.167	0.174
Fraction released from microparticles	0.299	0.281
Fraction bound to the microparticles	0.005	0.005
Measured concentration in all fractions	0.471	0.461
Theoretical concentration	0.5	0.5
Entrapment efficiency (%)	59.78	56.24
Net loss (%)	5.90	7.80

**Table 4 gels-08-00686-t004:** Kinetic analysis results of the dissolution profiles.

Formulation	Criteria	Zero Order	First Order	Higuchi	Korsmeyer–Peppas	Weibull
IGP20.Piperine_KRB	R^2^-adjusted	−0.5894	0.8211	0.4431	0.9894	0.9856
	AIC	85.9625	66.2664	76.1901	41.7018	43.6409
	MSC	−2.1729	0.0155	−1.0871	2.7449	2.5295
IGP200.Piperine_KRB	R^2^-adjusted	−0.0545	0.7450	0.7083	0.9762	0.9637
	AIC	81.6471	68.8267	69.2687	47.2805	52.7882
	MSC	−1.3086	0.1159	0.0668	2.5099	1.8980
IGP20.Piperine_TGS	R^2^-adjusted	−0.2424	0.8091	0.6334	0.9916	0.9879
	AIC	83.9562	66.3932	72.5389	38.2104	42.8303
	MSC	−1.6574	0.2941	−0.3888	3.4255	2.9122
IGP200.Piperine_TGS	R^2^-adjusted	0.3023	0.8401	0.8641	0.9772	0.9609
	AIC	78.6442	65.3663	63.8679	47.6297	51.9458
	MSC	−0.6865	0.7888	0.9553	2.7595	2.2799

R^2^-adjusted = adjusted coefficient of determination, AIC = akaike information criterion, and MSC = model selection criterion.

**Table 5 gels-08-00686-t005:** Summary of variables applied during the ionic gelation method to investigate the effect of chitosan and tripolyphosphate concentrations on particle size.

Formulation	Chitosan Concentration (mol/L)	Tripolyphosphate Concentration (mol/L)	Concentration (%*w*/*v*) Ratio (CS:TPP) *
1	0.005	0.017	0.409
2		0.013	0.524
3		0.01	0.699
4		0.006	1.048
5		0.005	1.395
6		0.003	2.097
7	0.002	0.006	0.524
8	0.002		0.524
9	0.003		0.699
10	0.004		0.839
11	0.005		1.048
12	0.01		2.097

* CS = chitosan; TPP = tripolyphosphate.

## Data Availability

No new data were created or analyzed in this study. Data sharing is not applicable to this article.

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
