# Peer review of "Formulation of Chitosan Microparticles for Enhanced Intranasal Macromolecular Compound Delivery: Factors That Influence Particle Size during Ionic Gelation"

_gels, 2022, doi:10.3390/gels8110686_

Round 1

Reviewer 1 Report

The aim of this study was to prepare and characterize chitosan microparticulate formulations containing macromolecular model compounds (FITC-dextran 4400, FD-4) and a bio-enhancer (piperine). In the first part of the paper, the factors influencing the microparticle size during ionic gelation of chitosan were investigated in order to select the delivery platform for the macromolecular drug delivery.  The authors put considerable work into the paper but unfortunately, the structure of the manuscript makes understanding the content difficult.

I propose to clearly define the purpose, which is to develop a microparticle platform for nasal delivery of macromolecules of required permeability. After that structure the experimental part in a way to be concise and compact.  The applied abbreviations should be defined where they appear first.  The authors should provide the original particle-size distribution curves and summarize the characteristic parameters of the curves in a Table. The curve fittings can be omitted since the primary purpose is the optimization of the particle sizes and not the curve fitting analysis. Based on the optimization of the particle size distribution of the delivery base, macromolecules are built into the particles. This part should be clearly separated in the Materials and Methos section. 

To sum up, the authors put a considerable amount of valuable work, which can be improved with well-structuring and condensing paper.

Reviewer 2 Report

In the present MS, formulation of chitosan microparticles for enhanced intranasal macromolecular compound delivery was investigated. Unfortunately, the selected model drug, FITC-dextran 4400, was not suitable. In fact, there are many macromolecular drugs which have been used in clinic. Selecting a suitable macromolecular drug which has been used in clinic is important for this research.

Reviewer 3 Report

The paper written by Weyers el all shows the preparation and characterization of some microparticulate formulations for nasal drug administration. Those formulations are based on chitosan, dextran and piperin. The paper is a complex study, well written and is present interest for readers. Considering this, i recommend publication the manuscript in Gels journal. 

Round 2

Reviewer 2 Report

Accept

Author Response

Dear Reviewer 2,

The manuscript has been extensively edited by a colleague in terms of English language and style. The formatting issues were also addressed, which arose due to an outdated version of Word.

Thank you for taking the time to review our manuscript and providing valuable feedback.